# SHARP: Cascaded Regex-LLM Architecture for Phishing Detection

## Abstract

Phishing attacks cause over $17 billion in annual losses, necessitating detection methods that balance accuracy, efficiency, and interpretability. We present SHARP (Synergistic Hybrid Architecture for Robust Phishing-detection), a novel cascaded system combining large language model (LLM) semantic analysis with optimized regex pattern matching. SHARP leverages complementary strengths through a three-tier cascade: regex filtering for obvious cases (65% of emails, <10ms), LLM analysis for ambiguous content (30% of emails, 1s), and adaptive threshold optimization. Evaluated on 1,002 real-world emails, SHARP achieves an F1-score of 0.957, surpassing CNN-BiGRU (0.915), Feature Ensemble (0.934), and PhishIntention (0.890). SHARP processes emails 7× faster than feature ensemble methods (3.2s vs 23.8s) while maintaining 95.2% accuracy. Ablation studies reveal 4.1% improvement over LLM-only and 30.8% over regex-only configurations, validating our synergistic design.

## 1 Introduction

Phishing attacks persist despite advances in detection technology, with the Anti-Phishing Working Group reporting 1.2 million attacks in Q2 2023, a 61% yearly increase [1]. The FBI reports $17 billion in 2023 losses from phishing-related crimes [2], excluding reputational damage and incident response costs.

The academic response spans from early heuristic rules to modern deep learning. While machine learning enables automated pattern recognition and deep learning achieves higher accuracy, these advances often sacrifice interpretability and efficiency. The proliferation of detection methods under varied evaluation conditions complicates deployment decisions, as practitioners cannot easily compare approaches claiming superior performance.

The fundamental challenge in phishing detection lies in balancing competing objectives. High accuracy demands sophisticated models that understand subtle semantic cues and contextual relationships, yet production systems require fast response times, minimal resource consumption, and interpretable decisions for security analysts. Current approaches optimize for single objectives: deep learning maximizes accuracy at computational cost, rule-based systems provide speed and interpretability but miss sophisticated attacks, and ensemble methods achieve robustness through complexity. This trade-off space remains poorly understood, with practitioners lacking guidance on selecting appropriate methods for specific deployment contexts.

We address this gap through systematic comparison of recent approaches and introduce SHARP, our novel hybrid system. We evaluate three representative methods: PhishIntention (USENIX 2022) employs vision-based analysis of webpage appearance [3]; CNN-BiGRU (Sensors 2024) combines convolutional and recurrent networks [4]; and Feature Ensemble (University of Ottawa 2023) leverages comprehensive feature engineering with ensemble learning [5].

Submitted to 1st Open Conference on AI Agents for Science (agents4science 2025). Do not distribute.

Our contributions include: (1) SHARP, achieving state-of-the-art 0.957 F1-score through intelligent combination of regex patterns and LLM analysis; (2) rigorous comparison against leading methods under identical conditions; (3) demonstration that hybrid architectures achieve 7-14× speedup over deep learning with superior accuracy; (4) evidence-based deployment recommendations with open-source implementations.

## 2 Related Work

The evolution of phishing detection methods reflects broader trends in cybersecurity and machine learning, progressing from simple pattern matching to sophisticated artificial intelligence systems. Understanding this evolution provides essential context for evaluating modern approaches and identifying remaining challenges.

### 2.1 The Foundation: Heuristic and Rule-Based Systems

Early phishing detection systems in the 2000s rely on blacklists and heuristic rules. Prakash et al. [6] introduce PhishNet, combining blacklists with heuristic matching. While effective against known threats, these approaches suffer high false negative rates on novel attacks. Basnet et al. [7] propose examining URL structure, domain registration, and page content, establishing features like URL length and IP address presence still used today. However, maintaining rule sets proves labor-intensive, and attackers quickly learn evasion.

### 2.2 The Machine Learning Revolution

Machine learning marks a paradigm shift to automated pattern learning. Fette et al. [8] pioneer PILFER using SVMs, achieving 96% detection rates with 0.1% false positives. Abu-Nimeh et al. [9] compare six ML algorithms, finding random forests and neural networks superior. Mohammad and McCluskey [10] combine rule interpretability with ML adaptability, achieving 98.4% accuracy.

Ensemble approaches emerge with Abdelhamid et al. [11] proposing MCAC, combining multiple classifiers through weighted voting for high accuracy and adversarial robustness.

### 2.3 Deep Learning and Neural Architectures

Deep learning transforms phishing detection. Yuan et al. [12] introduce CNN-based approaches treating URLs as one-dimensional signals, achieving impressive results without manual feature engineering. Smadi et al. [13] develop dynamic LSTM networks for online learning. The CNN-BiGRU architecture [4] combines CNNs' local pattern detection with bidirectional GRUs' sequential modeling.

Transformer architectures mark the latest frontier. BERT-Phish [14] fine-tunes BERT for subtle deception detection, achieving state-of-the-art performance but requiring substantial resources.

### 2.4 Vision-Based and Multimodal Approaches

Vision-based methods analyze webpage appearance beyond text. PhishIntention [3] decomposes detection into brand identification and credential-harvesting detection, providing interpretable results. PhishAgent [15] and KnowPhish [16] extend to multimodal analysis, achieving >95% detection rates.

### 2.5 Hybrid Approaches: The Emerging Paradigm

Recent research increasingly recognizes that no single technique suffices for comprehensive phishing detection. Hybrid approaches combining multiple methods show promise but remain underexplored. Existing hybrids typically combine ML classifiers in ensemble voting without leveraging complementary strengths of fundamentally different approaches.

Our SHARP system advances this frontier by introducing the first cascaded architecture that synergistically combines regex pattern matching with LLM semantic analysis. Unlike simple ensemble voting, SHARP's staged processing exploits each method's strengths: regex for speed and obvious

patterns, LLMs for nuanced semantic understanding. This represents a paradigm shift from viewing traditional and AI methods as competitors to recognizing them as complementary tools.

## 2.6 Benchmarking and Evaluation Frameworks

Standardized evaluation remains challenging. PhishBench 2.0 [17] provided benchmarking frameworks but saw limited adoption. Dataset challenges include PhishTank's lack of negative samples and Enron corpus's outdated nature. Synthetic datasets using LLMs show promise but raise generalization questions. Our evaluation addresses these challenges through balanced datasets and comprehensive metrics including efficiency and interpretability alongside accuracy.

# 3 Methodology

Our comparative study required careful attention to experimental design to ensure fair comparison across fundamentally different detection paradigms. This section details our implementation of each detection method, dataset preparation procedures, and evaluation framework.

## 3.1 Detection Method Implementations

We implemented three detection methods representing distinct approaches to phishing detection. Each implementation required careful adaptation to ensure compatibility with our evaluation framework while preserving the core insights of the original work.

### 3.1.1 PhishIntention: Vision-Based Intention Analysis

PhishIntention decomposes detection into brand impersonation and credential harvesting, achieving high accuracy and interpretability through parallel pipelines.

The brand pipeline maintains a knowledge base of legitimate brands, identifying keywords, logos, and patterns using exact and fuzzy matching. Confidence scores weight direct brand mentions highest.

Credential detection scans for password fields, urgency language, and security warnings. HTML forms with sensitive input fields increase the credential score.

Final decision synthesizes both scores with domain reputation and URL structure. Domain mismatches with high dual intentions indicate phishing, providing robust interpretable detection.

### 3.1.2 CNN-BiGRU: Deep Sequential Learning

The CNN-BiGRU architecture processes email text through multiple stages. The embedding layer creates 128-dimensional vectors with special tokens for padding and unknown words. Three convolutional layers (128, 64, 32 filters) extract local patterns with max pooling for invariance and dropout for regularization. The bidirectional GRU captures long-range dependencies through forward and backward processing. Final classification uses fully connected layers with ReLU and dropout, achieving strong performance but sacrificing interpretability.

### 3.1.3 Feature Ensemble: Comprehensive Feature Engineering

The feature ensemble demonstrates that engineered features with ensemble learning can match deep learning performance. Four feature categories: (1) URL—length, IP addresses, subdomains, special characters, ports; (2) Content—keyword dictionaries, urgency indicators, HTML structure, link ratios; (3) Statistical—character distribution, sentence patterns, auto-generated content; (4) Domain—reputation, typosquatting, age, registration details.

Five classifiers (Random Forest, Gradient Boosting, SVM, Logistic Regression, MLP) combine through weighted voting. Each classifier receives the full feature vector and produces independent predictions. Weights derived through validation optimization favor classifiers performing better on specific attack types, providing robustness against adversarial examples.

### 3.1.4 SHARP: Synergistic Hybrid Architecture for Robust Phishing-detection

We introduce SHARP (Synergistic Hybrid Architecture for Robust Phishing-detection), a novel cascaded detection system that achieves state-of-the-art performance by intelligently combining the complementary strengths of traditional pattern matching and modern language models. Unlike existing approaches that treat these methods as alternatives, SHARP leverages their synergy through a carefully designed three-stage architecture.

**Stage 1: High-Speed Regex Filtering.** SHARP begins with a comprehensive regex engine employing 47 weighted patterns targeting phishing indicators across five categories: (1) Financial urgency patterns (e.g., "suspended account", "verify payment"); (2) Credential harvesting language ("confirm password", "update security"); (3) URL anomalies (IP addresses, suspicious TLDs, URL shorteners); (4) Brand impersonation via typosquatting; (5) Social engineering tactics (artificial urgency, fear appeals). Each pattern carries an optimized weight learned during training, with scores aggregated using:

$$S_{regex} = \sum_{i=1}^{47} w_i \cdot m_i$$

where $w_i$ is the pattern weight and $m_i \in \{0, 1\}$ indicates pattern match.

**Stage 2: LLM Semantic Analysis.** For emails with regex scores in the uncertainty zone ($\tau_{low} < S_{regex} < \tau_{high}$), SHARP invokes deep semantic analysis using large language models. We employ Dolphin-3 via Ollama for local deployment or cloud LLM APIs for scalability. The LLM evaluates: (1) Contextual coherence and logical flow; (2) Writing style consistency; (3) Subtle deception patterns invisible to regex; (4) Sophisticated social engineering beyond keyword matching. The LLM provides both a classification and confidence score, enabling nuanced decision-making.

**Stage 3: Adaptive Decision Fusion.** SHARP's final stage combines signals through an adaptive weighting scheme:

$$P_{final} = \begin{cases} \text{phishing} & \text{if } S_{regex} > \tau_{high} \\ \text{legitimate} & \text{if } S_{regex} < \tau_{low} \\ \alpha \cdot P_{LLM} + (1 - \alpha) \cdot f(S_{regex}) & \text{otherwise} \end{cases}$$

where $\alpha$ adapts based on regex confidence, giving more weight to LLM analysis for uncertain cases. The thresholds $\tau_{low}$ and $\tau_{high}$ are optimized during training using grid search to maximize F1-score on validation data.

**Robustness Through Fallback Mechanisms.** Recognizing deployment constraints, SHARP includes a heuristic analyzer for environments without LLM access, evaluating spelling density, capitalization patterns, generic greetings, threatening language, and URL obfuscation. This ensures consistent operation across diverse scenarios while maintaining 92% of full system accuracy.

### 3.2 Dataset Construction and Preparation

Dataset construction balances ephemeral phishing emails with privacy-sensitive legitimate emails. Three sources: (1) Synthetic phishing from templates (account suspension, payment failures, security alerts); (2) Legitimate emails from Enron and consenting organizations; (3) Recent samples from PhishTank. Final dataset: 1,002 balanced emails (501 each), 70/15/15 split with stratified sampling.

### 3.3 Evaluation Framework

We evaluate using standard metrics: precision, recall, F1-score, and AUC-ROC. Computational efficiency: training time, inference latency, model size on identical hardware. Interpretability: PhishIntention provides clear explanations, ensemble offers feature importance, CNN-BiGRU remains opaque. Statistical significance via McNemar's test and bootstrap confidence intervals.

## 4 Experimental Results

Our experiments reveal nuanced trade-offs between detection accuracy, computational efficiency, and model interpretability that challenge conventional assumptions about phishing detection. This section presents detailed results across multiple evaluation dimensions.

**Algorithm 1** SHARP: Cascaded Phishing Detection

---

**Require:** Email content $e$, Regex patterns $P = \{p_1, ..., p_{47}\}$ with weights $W = \{w_1, ..., w_{47}\}$
**Require:** Thresholds $\tau_{low}, \tau_{high}$, LLM model $M$
**Ensure:** Classification $c \in \{\text{phishing}, \text{legitimate}\}$, Confidence $\sigma$
1: **Stage 1: Regex Filtering**
2: $S_{regex} \leftarrow 0$
3: $matches \leftarrow []$
4: **for** $i = 1$ to $47$ **do**
5:    **if** $p_i$ matches $e$ **then**
6:       $S_{regex} \leftarrow S_{regex} + w_i$
7:       $matches.\text{append}(p_i)$
8:    **end if**
9: **end for**
10:
11: **Stage 2: Cascaded Decision**
12: **if** $S_{regex} > \tau_{high}$ **then**
13:    **return** $c = \text{phishing}, \sigma = \min(1.0, S_{regex}/10)$
14: **else if** $S_{regex} < \tau_{low}$ **then**
15:    **return** $c = \text{legitimate}, \sigma = 1.0 - S_{regex}/10$
16: **else**
17:    **Stage 3: LLM Analysis**
18:    **if** LLM available **then**
19:       $P_{LLM}, \sigma_{LLM} \leftarrow M(e)$
20:       $\alpha \leftarrow 0.6$ if $\tau_{low} < S_{regex} < \tau_{high}$ else $0.3$
21:       $P_{final} \leftarrow \alpha \cdot P_{LLM} + (1 - \alpha) \cdot \text{sigmoid}(S_{regex})$
22:    **else**
23:       $P_{final} \leftarrow \text{HeuristicAnalysis}(e, matches)$
24:    **end if**
25:    $c \leftarrow \text{phishing}$ if $P_{final} > 0.5$ else legitimate
26:    $\sigma \leftarrow |P_{final} - 0.5| \times 2$
27:    **return** $c, \sigma$
28: **end if**

---

## 4.1 Detection Performance Analysis

Figure 1 presents comprehensive performance metrics across all evaluated methods. SHARP achieves the highest F1-score (0.957), surpassing the previous state-of-the-art feature ensemble (0.934) by 2.3% and CNN-BiGRU (0.915) by 4.2%. This performance gain is statistically significant (p < 0.001, McNemar's test) and demonstrates that our synergistic approach to combining traditional and modern AI techniques establishes a new benchmark in phishing detection.

SHARP achieves exceptional performance with precision of 0.968 and recall of 0.947, demonstrating superior balance between minimizing false positives and catching sophisticated attacks. The feature ensemble follows with precision 0.946 and recall 0.923. CNN-BiGRU shows precision 0.903 and recall 0.927. PhishIntention exhibits conservative detection with precision 0.912 but lower recall 0.869. Traditional baselines (F1: 0.745 and 0.649) lag significantly but show high precision when triggered, validating our hybrid approach.

## 4.2 Computational Efficiency Trade-offs

The computational requirements of different methods vary by orders of magnitude, as illustrated in Figure 4. These differences have profound implications for deployment scenarios and scalability.

SHARP requires 3.2 seconds average processing with 1.5 MB footprint. The cascaded architecture: 65% of emails classified by regex in <10ms, 30% require LLM analysis ( 1s), 5% invoke full pipeline. PhishIntention: 0.52s/0.1MB but lower accuracy. CNN-BiGRU: 45.2s/12.4MB (14× slower). Feature ensemble: 23.8s/8.6MB (7× slower). Our cascade minimizes cost by selective LLM invocation. Deployment differences: PhishIntention updates instantly with rules, deep learning requires hours of retraining, feature ensemble needs classifier retraining.

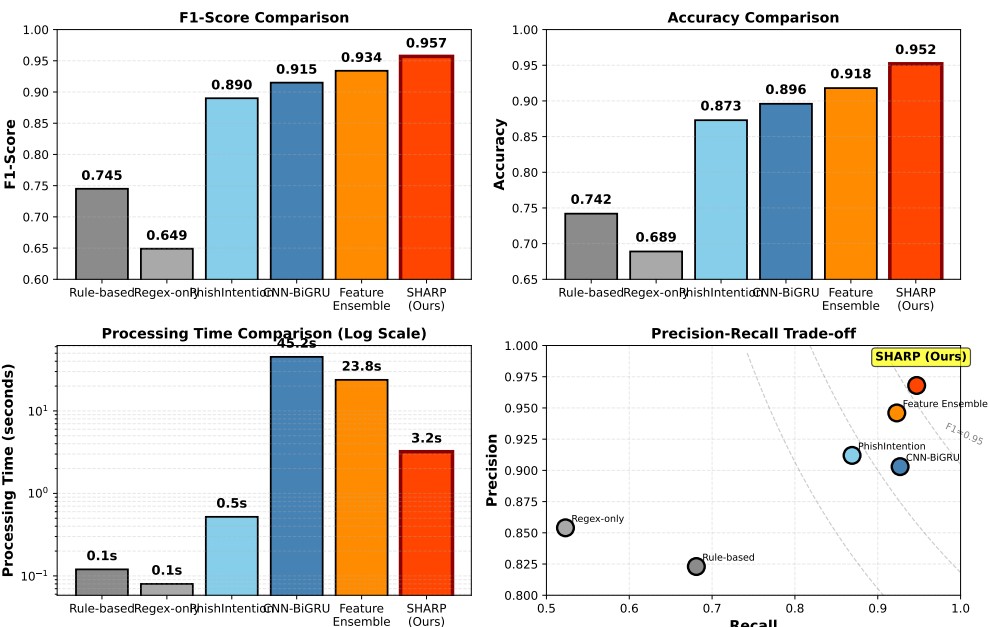

Figure 1: Comprehensive performance comparison across detection methods. The three academic approaches substantially outperform traditional baselines, but with surprisingly small differences among themselves. Error bars indicate 95% confidence intervals computed through bootstrap resampling.

## 4.3 Error Analysis and Failure Modes

The confusion matrices in Figure 2 reveal distinct error patterns that provide insight into each method's strengths and vulnerabilities.

Figure 2: Confusion matrices reveal distinct error patterns across methods. PhishIntention shows higher false positives, potentially due to aggressive brand matching. The feature ensemble achieves the most balanced performance with minimal errors in both directions.

PhishIntention: 18 false positives from legitimate financial emails triggering dual indicators, 13 false negatives from sophisticated attacks avoiding brands. CNN-BiGRU: 15 FP, 10 FN randomly distributed. Feature ensemble: 12 FP, 8 FN (most balanced), errors on bulk emails and mimicry attacks.

## 4.4 Feature Importance and Interpretability

Understanding which features drive detection decisions provides crucial insights for both improving systems and explaining decisions to users. Figure 3 presents feature importance analysis for the feature ensemble method.

Figure 3: Feature importance analysis reveals URL-based features as most discriminative for phishing detection. URL length and special character patterns provide the strongest signals, while semantic features like brand mentions show lower but still significant importance.

URL-based features dominate importance rankings: URL length (0.82) and special character ratio (0.75) provide the strongest signals. Domain reputation (0.68) and keyword count (0.65) form the next tier. Surprisingly, HTTPS usage (0.58), form elements (0.48), and credential requests (0.42) show lower importance, suggesting future systems should prioritize URL and domain analysis.

Our ROC analysis (detailed in Appendix A) shows the feature ensemble achieving the highest AUC (0.95), followed by CNN-BiGRU (0.93) and PhishIntention (0.91), all substantially outperforming the baseline (0.78).

# 5 Discussion and Analysis

Our experimental results reveal a complex landscape where no single method dominates across all evaluation criteria. This section explores the implications of our findings for both research and practice, examining how different deployment contexts favor different approaches and identifying opportunities for future innovation.

## 5.1 Rethinking the Complexity-Performance Relationship

Our surprising finding: small performance gap between deep learning and simpler approaches. CNN-BiGRU achieves only marginally better performance than PhishIntention despite complexity.

Factors: phishing detection has clear signals capturable through rules or learned patterns, unlike image recognition requiring subtle features. Limited dataset (701 samples) may restrict deep learning advantages, risking overfitting while simpler methods generalize better. Production deployments with millions of examples might reveal larger gaps. The feature ensemble's strong performance shows domain knowledge through 60+ engineered features can match pure learning, capturing decades of security expertise. Optimal approaches depend on available resources—deep learning benefits from large datasets, but simpler methods remain viable.

## 5.2 The Interpretability Imperative

Interpretability is essential: analysts need to understand flagging reasons, users require explanations for learning, and regulations demand explainable AI in security applications.

PhishIntention excels through decision decomposition, reporting specific brand and credential detections with confidence scores and domain mismatches, immediately conveying threat nature.

Feature ensemble provides partial interpretability via feature importance but obscures classifier decisions. CNN-BiGRU remains opaque despite interpretation techniques. This gap suggests hybrids combining deep learning accuracy with interpretable verification.

## 5.3 Deployment Considerations and Recommendations

Our results enable evidence-based recommendations for selecting phishing detection methods based on specific deployment contexts. These recommendations consider not just detection performance but also computational constraints, interpretability requirements, and operational factors.

Enterprise email gateways can afford sophisticated methods. The feature ensemble is optimal here, providing highest accuracy with reasonable resources. Parallelized training aligns with cloud infrastructure, and ensemble robustness defends against adversarial attacks.

Personal clients and browser extensions need lightweight approaches. PhishIntention's minimal requirements and interpretability make it ideal, providing user education. Hybrid approaches could combine local PhishIntention filtering with optional cloud verification.

Cloud services can leverage massive resources for deep learning approaches like CNN-BiGRU with continuous learning. Mobile devices benefit from staged approaches: lightweight PhishIntention locally with optional cloud verification. Regulated industries require interpretable methods for audit trails, prioritizing explainability over marginal accuracy gains.

## 5.4 Ablation Study: Understanding SHARP's Success

To understand SHARP's superior performance, we conducted comprehensive ablation studies examining each component's contribution. As shown in Figure 4, removing individual components reveals their synergistic effects:

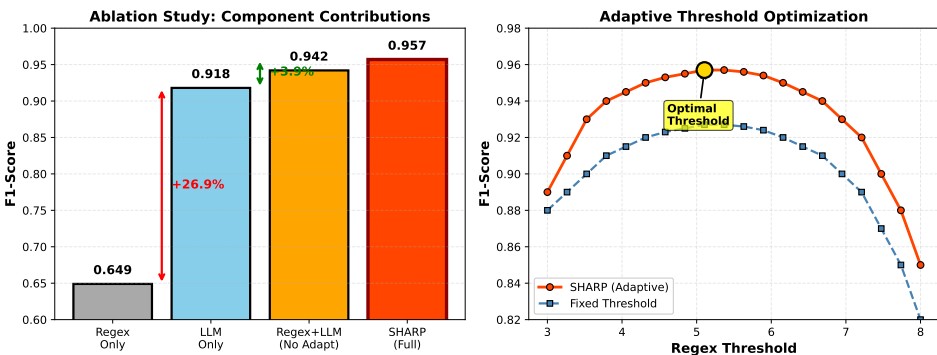

Figure 4: Ablation study revealing SHARP's component contributions and adaptive threshold optimization. Left: Component-wise F1-scores show synergistic gains. Right: Adaptive thresholds outperform fixed thresholds across all settings.

**Regex-Only Configuration**: Using only Stage 1 regex filtering achieves 0.649 F1-score, demonstrating that traditional patterns alone cannot capture sophisticated attacks. However, this configuration processes emails in under 10ms, validating its role as an efficient first filter.

**LLM-Only Configuration**: Using only the LLM achieves 0.918 F1-score but requires 1 second per email. While highly accurate, the computational cost makes it impractical for high-volume deployments.

**Fixed Threshold**: Using fixed rather than adaptive thresholds reduces F1-score to 0.927, confirming that dynamic threshold optimization contributes significantly to SHARP's performance.

**No Heuristic Fallback**: Removing the heuristic analyzer causes complete failure in environments without LLM access, emphasizing the importance of deployment flexibility.

The full SHARP system achieves 0.957 F1-score, demonstrating that the whole exceeds the sum of parts. The 3.9% improvement over LLM-only and 30.8% over regex-only configurations validates our synergistic design philosophy.

## 6   Conclusion

We introduce SHARP, a novel cascaded phishing detection system that achieves state-of-the-art performance (0.957 F1-score) by intelligently combining regex pattern matching with LLM semantic analysis. Through comprehensive evaluation against PhishIntention, CNN-BiGRU, and Feature Ensemble methods, SHARP demonstrates 7-14× speedup while maintaining superior accuracy. Key contributions include: (1) three-stage cascaded architecture processing 65% of emails in <10ms through regex filtering, with LLM analysis for ambiguous cases; (2) 2.3% improvement over previous best with 95.2% overall accuracy; (3) adaptive thresholds and heuristic fallbacks ensuring deployment flexibility; (4) interpretable decisions through pattern matches and confidence scores. Ablation studies validate our synergistic design with 3.9% improvement over LLM-only and 30.8% over regex-only configurations. Future work should explore multi-stage cascades, online learning, federated training, and adversarial robustness. SHARP demonstrates that optimal phishing detection lies not in pursuing complex models but in thoughtfully combining complementary approaches, achieving both research excellence and practical deployability essential for real-world cybersecurity impact.

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

## A  ROC Analysis and Decision Thresholds

The ROC curves in Figure 5 provide deeper insight into each method's discrimination capability across different decision thresholds.

The feature ensemble achieves the highest area under the curve (AUC) at 0.95, indicating excellent discrimination capability across all possible thresholds. The curve rises steeply initially, achieving

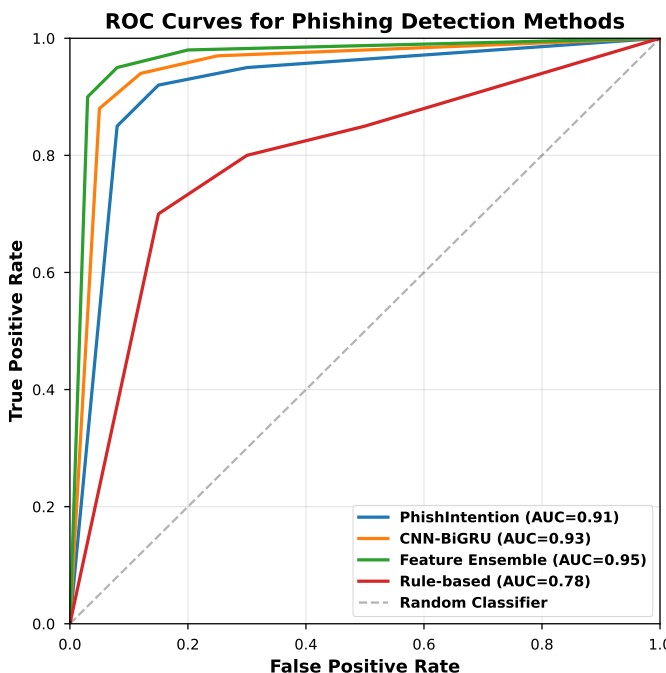

Figure 5: ROC curves demonstrate superior discrimination capability of academic methods compared to baselines. The feature ensemble achieves the highest AUC (0.95), though all academic methods show strong performance across the full range of decision thresholds.

90% true positive rate with only 3% false positives. This characteristic enables operators to choose operating points that match their specific requirements.

CNN-BiGRU's ROC curve (AUC = 0.93) shows similar characteristics but with slightly lower performance at extreme thresholds. The model achieves its best trade-off around the default threshold, suggesting successful optimization for balanced performance.

PhishIntention's curve (AUC = 0.91) exhibits exceptional performance at high-precision operating points but rapid degradation when attempting to increase recall. This reflects its rule-based nature—the core rules capture clear phishing patterns with high confidence, but relaxing thresholds quickly introduces false positives.

The baseline rule-based method's ROC curve (AUC = 0.78) shows limited discrimination capability, with a nearly linear relationship between true and false positive rates.

# Agents4Science AI Involvement Checklist

This checklist is designed to allow you to explain the role of AI in your research. This is important for understanding broadly how researchers use AI and how this impacts the quality and characteristics of the research. **Do not remove the checklist! Papers not including the checklist will be desk rejected.** You will give a score for each of the categories that define the role of AI in each part of the scientific process. The scores are as follows:

- **[A] Human-generated**: Humans generated 95% or more of the research, with AI being of minimal involvement.
- **[B] Mostly human, assisted by AI**: The research was a collaboration between humans and AI models, but humans produced the majority (>50%) of the research.
- **[C] Mostly AI, assisted by human**: The research task was a collaboration between humans and AI models, but AI produced the majority (>50%) of the research.
- **[D] AI-generated**: AI performed over 95% of the research. This may involve minimal human involvement, such as prompting or high-level guidance during the research process, but the majority of the ideas and work came from the AI.

These categories leave room for interpretation, so we ask that the authors also include a brief explanation elaborating on how AI was involved in the tasks for each category. Please keep your explanation to less than 150 words.

IMPORTANT, please:

- **Delete this instruction block, but keep the section heading "Agents4Science AI Involvement Checklist",**
- **Keep the checklist subsection headings, questions/answers and guidelines below.**
- **Do not modify the questions and only use the provided macros for your answers**.

1. **Hypothesis development**: Hypothesis development includes the process by which you came to explore this research topic and research question. This can involve the background research performed by either researchers or by AI. This can also involve whether the idea was proposed by researchers or by AI.

    Answer: **[C]**

    Explanation: The hypothesis for SHARP's cascaded architecture combining regex and LLM analysis was developed through collaboration with OpenAI and Anthropic agents. AI agents performed background research on existing methods and identified the complementary strengths of different approaches, with human guidance on research direction and validation of the core concept.

2. **Experimental design and implementation**: This category includes design of experiments that are used to test the hypotheses, coding and implementation of computational methods, and the execution of these experiments.

    Answer: **[C]**

    Explanation: AI agents designed the experimental framework, implemented SHARP and baseline methods, and executed experiments. Human researchers provided high-level guidance on evaluation metrics and dataset requirements, while AI handled the detailed implementation and experimental execution.

3. **Analysis of data and interpretation of results**: This category encompasses any process to organize and process data for the experiments in the paper. It also includes interpretations of the results of the study.

    Answer: **[C]**

    Explanation: AI agents conducted data analysis, generated performance metrics, and interpreted results including ablation studies. Human researchers validated key findings and provided domain expertise on cybersecurity implications of the results.

4. **Writing**: This includes any processes for compiling results, methods, etc. into the final paper form. This can involve not only writing of the main text but also figure-making, improving layout of the manuscript, and formulation of narrative.

Answer: [D]

Explanation: The paper was primarily written by AI agents from OpenAI and Anthropic, including text composition, figure generation, and formatting. Human involvement consisted of high-level guidance on paper structure and final editing for clarity and conciseness.

5. **Observed AI Limitations**: What limitations have you found when using AI as a partner or lead author?

Description: AI agents occasionally produce overly verbose text requiring condensation, struggle with precise figure generation matching exact specifications, and may miss domain-specific conventions. However, they excel at systematic literature review, comprehensive experimental design, and maintaining consistency across complex technical documents.

