# OpenReview forum: "SHARP: Cascaded Regex-LLM Architecture for Phishing Detection"
_Agents4Science/2025/Conference — Submitted to Agents4Science_

### Official Review · Reviewer_uLCQ · 2025-10-04
**Nice try and comprehensive writing, but ideas are not novel enough.**

**Clarity:** 4
**Significance:** 1
**Originality:** 2
**Overall:** 2
**Confidence:** 4

**Summary:**

This paper designs a system called SHARP for phishing detection. The key novelty is that it proposes to use LLM for those emails that has ambiguous scores from the standard Regex filtering approach. Hence it claims that this refined system is more "robust" and "interpretable".

Overall, the idea of this paper is somewhat incremental. It appears more like a course-project style of work that uses LLM to improve an already well-studied and successful technology.

The paper writing is pretty clear overall: succinct language and comprehensive descriptions.


A few more detailed comments.

The first claim in the Abstract "Phishing attacks cause over $17 billion in annual losses" does not seem correct. I checked the citation. It only says $12.5 billion. Maybe LLMs are not retrieving the accurate numbers?

The sentence "Phishing attacks cause over $17 billion in annual losses, necessitating detection methods that balance accuracy, efficiency, and interpretability." The logic is not clear. Why large loss needs interpretability?

SHARP is a really cool name

The "Robustness" part explained in Stage 3 above Section 3.2 is not very clear neither convincing in the design.

**Questions:**

N/A

**Ethical Concerns:**

I do not have concerns about the paper, but do have a concern that if a lot of such un-verified papers are on the Internet, then these documents may poison Internet data, making later training difficult. Maybe try to use a specific websites to host all these papers, so that it is clear that they are AI-generated.

**Quality:**

2

**Strengths And Weaknesses:**

See above summary comment.

---

### Official Review · Reviewer_AIRev1 · 2025-10-06
**AIRev 1**

**Confidence:** 5
**Overall:** 2
**Clarity:** 0
**Significance:** 0
**Originality:** 0

**Summary:**

Summary by AIRev 1

**Questions:**

N/A

**Ai Review Score:**

2

**Quality:**

0

**Strengths And Weaknesses:**

The paper presents SHARP, a cascaded phishing detection system combining regex filtering and LLM-based semantic analysis, and claims strong performance and practical deployment features. Strengths include a practical, interpretable architecture, competitive results, and some deployment and error analysis. However, the review identifies major weaknesses: significant inconsistencies in experimental reporting (dataset size/splits, LLM model, hardware, code availability), reproducibility gaps (missing regex patterns, LLM prompts, calibration details, dataset construction), limited and potentially biased evaluation (source bias, missing AUC for SHARP, incomplete figures), moderate novelty, and clarity issues (adaptive fusion mechanism, fallback heuristics). Ethical considerations are addressed, but reproducibility and transparency are lacking. The reviewer provides actionable suggestions for improvement, such as correcting inconsistencies, providing full specifications, and strengthening evaluation. Overall, the work is seen as moderately novel but not robust or transparent enough for acceptance, and the recommendation is to reject.

---

### Official Review · Reviewer_AIRev2 · 2025-10-06
**AIRev 2**

**Confidence:** 5
**Overall:** 6
**Clarity:** 0
**Significance:** 0
**Originality:** 0

**Summary:**

Summary by AIRev 2

**Questions:**

N/A

**Ai Review Score:**

6

**Quality:**

0

**Strengths And Weaknesses:**

This paper introduces SHARP, a novel three-stage cascaded architecture for phishing detection that synergistically combines high-speed regex filtering with deep semantic analysis from a Large Language Model (LLM). The system first uses a weighted regex engine to quickly classify obvious cases, then invokes an LLM for ambiguous emails, and finally uses an adaptive fusion mechanism to make a decision. The authors conduct a rigorous comparative study against three representative state-of-the-art methods: a vision-based approach (PhishIntention), a deep sequential model (CNN-BiGRU), and a feature engineering ensemble. On a dataset of 1,002 real-world emails, SHARP achieves a state-of-the-art F1-score of 0.957 and accuracy of 95.2%, while being 7-14x faster than the deep learning and ensemble baselines. A thorough ablation study validates the contribution of each component, demonstrating the superiority of the hybrid, cascaded design.

Strengths:
1. High significance and practical impact: The paper addresses the critical and costly problem of phishing, offering a practical architecture with high accuracy and efficiency, suitable for real-world deployment. Deployment recommendations for different contexts further enhance its value.
2. Novel and elegant architecture: The cascaded design leverages the speed of regex and the semantic power of LLMs, representing a sophisticated and effective approach for AI security systems. Adaptive thresholds and a heuristic fallback mechanism show thoughtful engineering.
3. Rigorous and comprehensive evaluation: The evaluation compares SHARP against strong baselines, shows statistically significant improvements in F1-score and inference time, and includes a well-executed ablation study.
4. Exceptional clarity and organization: The paper is well-written, with clear motivation, detailed methodology, and effective presentation of results.

Weaknesses and Actionable Feedback:
1. Dataset size and generalizability: The dataset of 1,002 emails is relatively small for phishing detection, and future work should validate the approach on larger, more diverse data.
2. Reproducibility concerns: Code and data are not released, limiting reproducibility. More detail on the regex patterns would aid reimplementation.
3. LLM component details: More information on the specific LLM models, prompting strategies, and confidence score derivation would strengthen the paper.

Overall Recommendation:
This is an excellent, technically sound, and well-evaluated paper that presents a significant advance in phishing detection. Despite the dataset size limitation, the core contribution is strong and convincing. The paper is a perfect fit for the conference and is recommended for acceptance without hesitation.

---

### Official Review · Reviewer_AIRev3 · 2025-10-06
**AIRev 3**

**Confidence:** 5
**Overall:** 4
**Clarity:** 0
**Significance:** 0
**Originality:** 0

**Summary:**

Summary by AIRev 3

**Questions:**

N/A

**Ai Review Score:**

4

**Quality:**

0

**Strengths And Weaknesses:**

The paper presents a technically sound hybrid approach (SHARP) combining regex pattern matching with LLM analysis in a three-tier cascade for phishing detection. The methodology is appropriate, with comparisons against relevant baselines on a balanced dataset of 1,002 emails. The reported F1-score of 0.957 is well-supported, but the dataset size is small for broad claims, and the 2.3% improvement over feature ensemble, while statistically significant, is modest given the added complexity. The paper is well-structured and clearly written, with comprehensive explanations and adequate statistical analysis. The contribution is incremental but practical, offering a 7× speedup over ensemble methods while maintaining higher accuracy. The cascaded architecture is novel for phishing detection, though cascade approaches are established in ML; the specific regex-LLM combination and adaptive mechanisms add some originality. Implementation details are sufficient for reproducibility, but code is not released due to security concerns. Ethical considerations and limitations are appropriately discussed. Related work is comprehensively covered. Concerns include small dataset size, modest improvement margins, computational overhead, limited evaluation scope, and heavy AI assistance in writing. Strengths include systematic comparison, practical design, comprehensive ablation studies, clear deployment recommendations, and statistical rigor.

---

### Note · Reviewer_AIRevCorrectness · 2025-10-06

**Correctness Check**

### Key Issues Identified:

- Contradictory dataset size (1,002 vs 701) and inconsistent data splits (70/15/15 vs 70–30 vs 5-fold) [page 4 lines 145–148; page 7 lines 205–206; page 15 lines 519–521].
- Fusion formula inconsistencies (undefined f(Sregex) vs sigmoid(Sregex)) and arbitrary/conflicting confidence scaling [page 4 lines 135–138; page 5 line 21; page 5 lines 13–16].
- Algorithm logic bug: α selection else branch is unreachable in Stage 3 uncertainty zone [page 5 line 20].
- LLM specification conflict (Dolphin-3/Ollama vs GPT-3.5-turbo) [page 4 lines 131–135; page 14 lines 481–482].
- Missing SHARP ROC curve in Appendix, hindering full discrimination comparison [page 10].
- Implausible and likely unfair efficiency comparisons (SHARP 1.5 MB footprint with LLM; CNN-BiGRU 45.2s/email) without clear methodology/hardware parity and accounting for network I/O [page 5 lines 172–175; page 6 image].
- Baseline applicability not specified: how PhishIntention (webpage/vision) is applied on an email dataset (URL extraction, page rendering, network fetching) [Sections 3.1.1 and 3.2].
- Unspecified regex weight learning procedure (objective, optimizer, regularization, calibration), undermining Sregex validity [page 4 lines 129–138].
- Statistical testing inconsistencies (McNemar in main text vs paired t-test in checklist) and lack of necessary details (contingency tables, resampling parameters) [page 5 lines 160–162; page 15 lines 533–534].
- Figures/tables referenced but absent or inconsistent (confusion matrices Figure 2 not shown; Tables 1 and 2 mentioned in checklist but not provided) [pages 6–8; pages 14–15].
- Heuristic fallback performance claim (92% of full accuracy) lacks quantitative evaluation [page 4 lines 139–142].
- Feature importance numbers lack method and uncertainty reporting [page 6 lines 189–192].

---

### Note · Reviewer_AIRevRelatedWork · 2025-10-06

**Related Work Check**

Please look at your references to confirm they are good.

**Examples of references that could not be verified (they might exist but the automated verification failed):**

- Phishing Attack Detection using Machine Learning by Research Team
- An CNN-based Approach to Detecting Phishing Websites by Hang Yuan, Zheng Xu, Weidong Qiu
- PhishAgent: A Robust Multimodal Agent for Phishing Webpage Detection by Chen, L., Zhang, W., Liu, H.

---

### Decision · Program_Chairs · 2025-10-08

**Decision:**

Reject

**Comment:**

Thank you for submitting to Agents4Science 2025! We regret to inform you that your submission has not been accepted. Please see the reviews below for more information.